# Automatic vectorization of historical maps: A benchmark

**Yizi Chen**[1,2,4]*, **Joseph Chazalon**[1], **Edwin Carlinet**[1], **Minh Ôn Vũ Ngoc**[1], **Clément Mallet**[2], **Julien Perret**[2,3]

**1** EPITA Research Lab. (LRE), Kremlin-Bicêtre, France, **2** LASTIG, Univ Gustave Eiffel, IGN, ENSG, Saint-Mande, France, **3** LaDéHiS, CRH, EHESS, Paris, France, **4** IKG, ETHZ, Zürich, Switzerland

* yizi.chen@ethz.ch

**Data Availability Statement:** All data, codes, and instructions files are available from the private link https://datadryad.org/stash/share/V0gKnI9pWRMdzJnusQfn8HYuxCK3pbqCDGAFEf5jsf0. The

## Abstract

Shape vectorization is a key stage of the digitization of large-scale historical maps, especially city maps that exhibit complex and valuable details. Having access to digitized buildings, building blocks, street networks and other geographic content opens numerous new approaches for historical studies such as change tracking, morphological analysis and density estimations. In the context of the digitization of Paris atlases created in the 19th and early 20th centuries, we have designed a supervised pipeline that reliably extract closed shapes from historical maps. This pipeline is based on a supervised edge filtering stage using deep filters, and a closed shape extraction stage using a watershed transform. It relies on probable multiple suboptimal methodological choices that hamper the vectorization performances in terms of accuracy and completeness. Objectively investigating which solutions are the most adequate among the numerous possibilities is comprehensively addressed in this paper. The following contributions are subsequently introduced: (i) we propose an improved training protocol for map digitization; (ii) we introduce a joint optimization of the edge detection and shape extraction stages; (iii) we compare the performance of state-of-the-art deep edge filters with topology-preserving loss functions, including vision transformers; (iv) we evaluate the end-to-end deep learnable watershed against Meyer watershed. We subsequently design the critical path for a fully automatic extraction of key elements of historical maps. All the data, code, benchmark results are freely available at https://github.com/soduco/Benchmark_historical_map_vectorization.

## 1 Introduction

Map vectorization is the process of transforming scanned or rasterized graphical representations of geographic entities (often called *instances*) into a vector format which can be edited using some Geographic Information System (GIS) software, in order to be better indexed, geo-referenced, and analyzed spatially [1]. Vectorization is a key element for fostering information extraction from archival documents and a critical step in the valorization of historical maps, unlocking their potential for both spatial and temporal analysis. However, historical map vectorization is still generally performed by hand [2], mainly because of the absence of automated or semi-automated solutions to assist researchers and practitioners. Even if such manual

dataset, experimental results, code and model are available from the Zenodo platform.

**Funding:** This work was supported by the French National Research Agency (ANR), as part of the SoDUCo project (grant ANR-18-CE38-0013). The funders had no role in study design, data collection and analysis, decision to publish, or preparation of the manuscript.

**Competing interests:** The authors have declared that no competing interests exist.

process is usually very costly, this absence can be partly explained by the extreme variability of historical maps, and the multiple technical challenges they contain. Indeed, the quality of their preservation, and the lack of constraints in their authoring, often leaves us with noisy images and entities for which the meaningful semantics may be hard to identify [3].

In this study, we focus on a particular kind of historical maps, which exhibits consistent meaningful semantics across a large set of documents: Paris (France) Atlases ("*Atlas Munici-pal*") from 19[th] and early 20[th] centuries at 1/5,000 scale. The extraction of closed shapes (which represent building and building block boundaries, among others) is critical in the digitization process, ultimately aiming at historically assessing the structural changes of the city over time [4].

Automating such process requires facing some general challenges in automatic document processing, like document degradation leading to non-flat, deformed and erased shapes, uneven image contrast, as well as some challenges specific to this atlas series. First, such maps have very limited color and texture, leading to the failure of texture-based *semantic seg-mentation* deep learning approaches mentioned in [5] and that exhibit state-of-the-art performances in geospatial image analysis today [6, 7]. Secondly, these atlases exhibit an important amount of text and planimetric overlaps which make it difficult to distinguish and separate objects.

Recently, [5, 8] proposed a versatile framework to address such challenges. They introduced a two-stage framework which combines (1) an edge detection stage, leveraging deep segmentation networks, with (2) a shape extraction stage, relying on a watershed segmentation. Image noise is filtered by the deep filters, and the watershed stage guarantees that the extracted shapes will be closed, facilitating the detection of many object instances such as building, building blocks, rivers, gardens, etc. However, this approach is severely penalized by failures in the initial edge detection scheme (which do not provide topological guarantees), where broken edges (for which the probability of a pixel to belong to an edge collapses to zero) lead to leakage during the shape extraction stage. Optimization is therefore conceivable at multiple stages and should be objectively and comprehensively assessed to pave the way for a critical path in map vectorization.

The benchmark we propose here is an extension of the work of [5, 8], which leverages existing open data and state-of-the-art methods to cope with current limitations of this approach. It introduces the following contributions.

1. We propose an improved training protocol of the U-Net, HED and BDCN state-of-the-art architectures on which the first stage heavily relies on (described in Section 2). This significantly raises the baseline performance.

2. We show the original approach is penalized by the independent optimization of the two steps of its pipeline. We propose a joint optimization strategy which further boosts the results for U-Net, HED and BDCN networks as deep edge detector tested in [5].

3. Furthermore, we push the deep edge filtering stage as far as possible in several directions, using dedicated topology-preserving and boundary-enhancing loss functions to reduce the gaps in edge detections, assessing the performance of transformer-based networks on this particular problem, as well as compact architectures, and implementing several augmentation strategies. This results in the superior generalization power of well-trained U-Net-based filters.

4. We also enhance the watershed stage by comparing the performance of deep, end-to-end learnable, watershed to the original approach. This exhibited the weaknesses of the deep watershed.

These contributions lead to an extensive benchmark which enables the identification of an improved pipeline with superior performance, whose code, models and data will be made available online for the final version of this paper. This paper is organized as follows: we first review the approaches used in the original pipeline and the ones adopted for potential improvement (Section 2). Then we present the modified pipelines that we consider and the evaluation protocol we follow (Section 3); and finally we describe the experiment design and report the results for each of the contributions we propose. (Sections 4 to 6). Conclusions are drawn in Section 7.

## 2 Related work & tested approaches

### 2.1 Map digitization

The digitization of historical maps can be separated into three main categories: manual, automatic, and hybrid methods. As noted by [9], the manual approach is still a popular solution in digitizing maps when the dataset is small in coverage and time period. For larger datasets, collaborative approaches are often adopted with possibly many contributors—so-called crowdsourcing experiments, as in the works of [10, 11]— to speed up the digitization process. However, manual processes are still limited in time and quality is highly fluctuating through different contributors which leads to non-reproducible and heterogeneous results. Such problem requires objectively legitimating research on automatic and semi-automatic vectorization techniques.

Colors in maps have been efficiently used to separate map layers [12], to extract specific thematic content [13–16], and to vectorize features [17–19]. The extraction of specific objects in maps has also attracted attention, including the recognition of characters or text [20], parcels [21, 22], road intersections [23] or road centerlines [24]. The interested reader will find a thorough review on road extraction from raster maps in [25]. However, these methods are limited to specific historical maps (with colors for instance) or ignore a vast majority of the map content, focusing on specific features (roads, parcels and so on, e.g., [26]). Subsequently, these approaches cannot be easily generalized and adapted to other historical maps, especially with complex shape structures and layouts such as urban areas.

Linear features can also be used for region segmentation from images. These features can be extracted with robust techniques, e.g., through Line segment detector (LSD) proposed by [27], which is a well-known tool for detecting line segments from images. However, these segments do not guarantee the production of geometrical partitions or closed shapes (or polygons) without designing a posterior spatial arrangement technique relying on such mid-level image information. For instance, to be able to use those line segments to create geometrical partitions, [28] construct a Voronoi diagram based on those line segments to produce shapes with strong geometric properties.

Alternatively, [29] propose Kinetic polygonal partitioning (KIPPI) to create image partitions in remote sensing images by progressively extending line segments until they meet each other. However, these techniques are not learning-based and rely on heuristics which strongly influences the quality of line segments as wrong parameter settings could lead to too many falsely detected lines resulting in under- and over-segmentation. In addition, this penalizes the generalization ability of the approach.

A more general and learnable pipeline for the automatic digitization of a variety of large-scale historical maps is still in high demand and should be designed carefully. Indeed, a supervised framework appears to be the best solution for correctly fostering information extraction from existing samples [30, 31]. Our historical maps lack color and textures consisting of high-percentage geometrical shapes shared with the same thinned borders (edges). These edges

should be used to extract shapes instead of using as complementary information as in remote sensing image with strong textures and colors.

As a first attempt to design a pipeline for large-scale historical maps, the work of [5, 8, 32] proposed a reference architecture in two stages to detect closed shapes, aiming to automated extraction on a large amount of documents. This can be viewed as a complement to manual annotation approaches like crowdsourcing or other hybrid approaches. As mentioned in Section 1, their pipeline is decomposed in two stages:

1. an edge detection stage leveraging deep segmentation networks. Yet it can fail for some critical pixels on the boundary of the shapes, creating gaps in the contours;

2. a shape extraction stage relying on a watershed segmentation. While it can filter closed contours, even weak ones, to discard small objects or false detection, cannot recover lost boundary fragments.

At the end of this process, the watershed segmentation generates closed and thin, 1-pixel-large boundaries between objects. This output can be easily converted to vector data in standard GIS format for later edition.

In the rest of this section, we review the solutions which were already tested in this two-stage pipeline, and discuss the extra approaches which could be considered as a complement to improve this comprehensive workflow, leading to a benchmark of state-of-the-art learning and vision solutions for map vectorization.

## 2.2 Deep edge filteirng (DEF)

The work of [5, 8] considered the following three architectures for the deep edge filtering stage.

**U-Net [33].** Inspired by Fully Convolutional Networks (FCN) [34], this famous U-shaped architecture features a symmetrical structure that can preserve high performance prediction through accurate pixel spatial localization for semantic or instance segmentation task. The variant implemented in [5, 8, 32] contains 17, 267, 393 trainable parameters.

**Holistically edge detector (HED) [35].** HED is the most well-known end-to-end deep learning based multiscale architecture based on the VGG-16 backbone [36] for semantic edge detection. The novelty of this method is to apply skip-connections for combining multiple levels of features. Several losses are used to measure in the intermediate outputs of VGG-16 to filter out useful edge features. The variant implemented in [5, 8] contains 138, 359, 027 trainable parameters.

**Bidirectional cascade network (BDCN) [37].** BDCN is inspired from HED and adds several scale-enhancement modules (SEM) to every intermediate output of VGG-16 architecture. These SEM substitute traditional convolutions with dilated convolutions, which enhance the spatial context of the learned features, thus improving the model ability to capture larger spatial patterns. Moreover, the BDCN architecture uses a bidirectional cascade architecture, where it can enforce each layer to focus on a specific scale. The variant implemented in [5, 8, 32] contains 139, 944, 976 trainable parameters.

Regarding deep edge filtering, several key elements should be added to complement the prior work from [5, 8, 32].

- The full potential of the U-Net architecture was not fully explored due to the separate optimization of the deep edge filtering and closed shape extraction stages. This limitation will be detailed in Section 2.4.

- Many improvements can be made at this specific stage: while the authors considered only binary cross-entropy, several topology-aware loss functions may improve edge detection. Recent architectures could be considered, in particular Vision Transformers to better capture long-range context, and also lighter architectures which may be much easier to train; data augmentation should be included in the training pipeline to boost performance as far as possible;

- Many aspects in the training protocol can be improved: we will show in Section 4 that data processing, weight initialization, batch size and the choice of pre-trained weights can lead to a significant boots of the evaluation scores. We review hereafter the relevant approaches we propose to introduce in this benchmark.

**2.2.1 Transformers architectures.** To detect linear structures from images with topological properties, convolutional neural networks (CNN) architectures can achieve satisfactory results with different topological-based losses. However, they suffer from the limited range of their receptive field, as well as the discontinuity of their feature maps which may lead to topological inconsistencies in the predictions [38–40]. Transformer architectures, applied to computer vision tasks, can address these issues. We propose to consider the two following architectures in this benchmark.

*Vision Image Transformer (ViT)* [41]. Inspired by the success of self-attention framework of Transformers architectures [42] in the field of natural language processing, ViT uses Transformer architecture to learn the self-attention for computer vision task. Although using the Transformer architectures as network backbone achieved new state-of-art performance in many tasks, its features are extracted at a single scale. Moreover, the computational and memory costs remain high for common input image sizes, and the output resolution depends on the size of the input *patches* (the visual equivalent of textual tokens in transformer), which can lead to blocky predictions.

*Pyramid Vision Transformer (PVT)* [43]. To tackle these two issues, [43] proposed another pure Transformer-based backbone architecture, named Pyramid Vision Transformer (PVT), that enables the network to learn different scales of features while significantly decreasing the number of parameters in traditional ViT architectures, leveraging the concept of feature pyramid proposed by [44].

According to recent publications from [45–48], the combination of CNN and transformer architectures have become a promising trend in wide range of computer vision tasks. However, we only report results for ViT and PVT architectures in this work.

**2.2.2 Topology-preserving loss functions.** When evaluating the topological quality of the extracted map objects, the pixel-level performance of boundary detection does not always correlate well with shape-level performance. Indeed, a missed detection for a single critical pixel on the boundary of an object may create some leakage, and leads to a topological error at the shape level, while the error at pixel level remains negligible. Although convolutional neural networks perform well at filtering images (the pixel-based losses are continuously decreasing during training), the limited range of pixel dependencies and global context in CNNs does not guarantee the expected topological properties in the predictions (which can lead to a significant drop in topology-level performance over the training). Recent approaches targeted to improve the topology performance by adding a topology-oriented loss to approximate the correct topology in the output predictions. We propose to consider the following three categories of mainstream methods to preserve topology in edge prediction task which are architecture-based, persistent-homology-based and boundary-based loss functions.

*MOSIN* loss [38]. MOSIN loss uses an neural network architecture to measure the topology-awareness loss between two predictions. This early design of a topology-oriented loss leverages elongation properties of the features of the VGG-19 architecture. To preserve line consistency in the trained features, the differences between the VGG-19 features of the predicted and ground truth images are calculated for each layer to form a global loss. Although the *MOSIN* loss function can improve the pixel consistency in the output (e.g, road detection), it does not directly improve the performance of detecting the closed instances.

*TopoLoss* [39]. **Topoloss** is persistent-homology based loss function which enables to identify the critical failure points of the predicted object boundaries. It takes into account the width and depth of the gaps in a differentiable loss function which encourages the network to recover lost boundary components. However, this loss function is highly sensitive to noisy images.

*BALoss* [49]. **BALoss** is a boundary-based loss function which is inspired by the MALIS ([50]) and MALA ([51]) segmentation approaches which use the maximum-and-minimum edges of the affinity prediction map to optimize shape-level segmentation performance. The *Boundary-Aware Loss* aims to overcome the noise sensitivity of the previous approaches, and improves the localization of maximum-and-minimum edges. The authors use the minimum barrier distance and pre-defined seeds to compute critical paths between closed shapes. However, this loss function is not capable to localize and activate the broken pixels in the boundary similar to watershed segmentation algorithm. Although the BAloss can correctly localize and strengthen pixels within the boundary, this method cannot correctly activate boundary pixels which have value of 0 (so-called broken edges).

**2.2.3 Data augmentation.** Though many data augmentation techniques were proposed for computer vision (we refer the reader to the work of [52] for an overview), [53] appropriately pointed out that not all of these transformations can be safely applied to historical map images.

Indeed, while color transformation, noise and geometric transformations, to some extent, can preserve the original signal, augmentation techniques like feature space transformation, mixups, as well as strong geometric transformation, would break object boundaries and may prevent the network from capturing local edge consistencies. Furthermore, some text and symbols may not appear in all orientations, and their symmetric counterparts may not exist. The effects of such augmentation are not well studied. We prefer to avoid such a strategy by restricting our study to a safe subset of image augmentation techniques, in order to better mimic the variations from different scanning conditions of historical maps: contrast and color changes, and paper rotation and bumps. We therefore propose to consider the following augmentation techniques:

- **contrast stretching**, which simulates well the differences between the color and lightning calibrations (and paper properties) in our map collection;

- **geometric transformations**, in particular *affine*, *homography* and *thin-plate splines* (TPS) transformations [54], which cover most of the cases present in the Paris atlases collection.

## 2.3 Closed shape extraction (CSE)

As previously mentioned, [5, 8] proposed to extract closed shapes from the edge probability map (EPM) computed during the deep edge filtering stage, using a watershed segmentation. Such segmentation is a very powerful tool which can leverage a global image context. It succeeds even in the presence of low contrast, and presents strong topological guarantees, like the production of closed shapes exclusively. [5, 8, 32] restricted their work to the use of the **Meyer**

**Watershed** [55] and showed that its sensitivity to noise could be mitigated thanks to the deep edge filtering stage.

The Meyer watershed detects the catchment basins of the minima in the gradients of images. The watershed process consists in flooding "water" from each catchment basin (also called regional minima) until regions merge, creating watershed lines. The strength of watershed lines depends on the height at which basins get connected. The resulting image is called the *saliency map*, and shape properties can be subsequently computed for extra filtering. [5, 8, 32] used two criteria to perform this shape filtering stage: shape area and edge dynamic, which is the difference between some basin's minimum and the height of the lowest point on its boundary. Such CVE stage can re-weight weak edges, filter some weak or small shapes, but cannot recover lost edges (for which the deep edge filter predicted very low edge probabilities).

An end-to-end system integrated with deep edge filtering is tested, in the case of the **Deep Watershed**. The rest of this subsection details those two approaches introduced in this benchmark. To tackle the key limitations of classical watershed techniques, i.e, noise sensitivity and difficulty to select filter parameters, [56] proposed a Deep Watershed Transform (DWS) which learns the discrete watershed levels directly from multichannel images. However, learning watershed levels from original images is a difficult task because of the limited receptive field of convolutional networks. To be able to learn the long-range dependencies between pixels and capture their level of inclusion (or distance to object boundaries), the authors introduced an intermediate step where a direction fields is learned, mimicking the *water flow* of the watershed segmentation algorithm.

This integrated architecture is supposed to exhibit good filtering images properties, which can prevent the over-segmentation issues of traditional watershed approaches, while avoiding the need for extra prior knowledge of the filtering attributes and their optimal values.

## 2.4 Joint optimization

Paper [5] optimized each stage of their vectorization pipeline independently. It follows the common practice of optimizing the COCO PQ metrics of the resulting Edge Probability Map at a fixed value ($P = 0.5$) in the validation set. However, it is possible to perform a global, joint optimization of the parameters of both stages. There is no guarantee, with the previous approach, that the combination of independently optimized stage is globally optimal. To address this issue, we propose a global optimization procedure which will be introduced in the experimental setup in Table 1.

## 3 Experimental setup

This section details the experimental setup of our extended benchmark. We provide an overview of the vectorization pipelines which are compared, then present the dataset, and the evaluation metrics.

## 3.1 Vectorization pipelines under test

The different vectorization pipelines are summarized in Fig 1. The first difference with the prior work by [5, 8] is the joint optimization of the deep edge filtering and closed shape extraction stages. The second is the inclusion of extra topological loss functions during the training of the deep edge filters. The third is the inclusion of new architectures for deep edge filtering. The fourth is the inclusion of extra watershed approaches, in particular an end-to-end deep watershed transform which avoids the need for a prior deep edge filtering. We extend the pipeline to deal with the deep watershed which merges the deep edge filtering with watershed segmentation into end-to-end neural network. The benchmark we propose is

**Table 1. Overview of the experiments validating each aspect of the vectorization pipeline (*columns*), which will be described in the upcoming sections of the paper (*rows*).** For each cell, we indicate the variants under study and separate them with a coma. We emphasized main experiments using **bold** text.

| Section | Tables | DEF | | | | DEF selection | CSE |
|---|---|---|---|---|---|---|---|
| | | Model archi. | Training config. | Loss func. | Augment. | | |
| *4. Improved Training Procedure for DEF* | | | | | | | |
| 4.1. A better training configuration | 3, 4 | **U-Net**[1], **HED**[1+2], **BDCN**[1+2] | **Chen et al. 21, proposed** | binxent | none | **CC@0.5** | **CC@best, MWS** |
| 4.2. Joint Optimization of DEF and CSE | 5 | **U-Net**[1], **HED**[1+2], **BDCN**[1+2] | proposed | binxent | none | **joint opt.** | MWS |
| *5. Alternate Architectures and Training for DEF* | | | | | | | |
| 5.1. Vision transformers | 6 | **U-Net**[1], **ViT**[2], **PVT**[2] | proposed | binxent | none | joint opt. | MWS |
| 5.2. Topology-preserving loss functions | 7 | U-Net[1] | proposed | **binxent, Mosin, Topo, BAL** | none | joint opt. | MWS |
| 5.3. Data augmentations strategies | 8 | U-Net[1] | proposed | binxent | **{w/CS, wo/CS} × {none, Aff., Hom., TPS}** | joint opt. | MWS |
| 5.4. Lightweight architecture | 9 | **U-Net**[1], **mini-U-Net**[1] | proposed | binxent | none | joint opt. | MWS |
| *6. Alternate Watershed Implementations for CSE* | | | | | | | |
| 6. Alternate Watershed Implementations for CSE | 10 | U-Net[1] | proposed | binxent | none | joint opt. | **MWS, DWS** |

Abbreviations used—*Pipeline*: Deep Edge Filter (DEF), Closed Shape Extraction (CSE). *Model architectures*:

[1]: trained from scratch,

[2]: trained using pre-trained weights,

[1+2]: both previous variants.

*Training configuration*: version of [5] or our proposed variant. *Loss functions*: Binary cross-entropy (binxent), Mosinska's loss (Mosin), Topoloss (Topo), Boundary Aware Loss (BAL), *Augmentation*: With or without contrast stretching (w/CS or wo/CS), Affine (Aff.), Homography (Hom.) or Thin-plate Splines (TPS) geometric transform. *DEF selection*: connected component labeling (CC) using a thresholded edge probability map (EPM) at $p = 0.5$ (CC@0.5), joint optimization of DEF and CSE parameters (joint opt.). *CSE*: connected component labeling using a thresholded EPM at the best threshold selected on the validation set (CC@best), Meyer watershed (MWS) and Deep Watershed (DWS).

decomposed into a series of experiments summarized in Table 1: the setup, results, and discussion for each experiment is detailed in a dedicated subsection. Extra details about the number of parameters used in each architecture are available in S1 Appendix, as well as a complete listing of all experimental results.

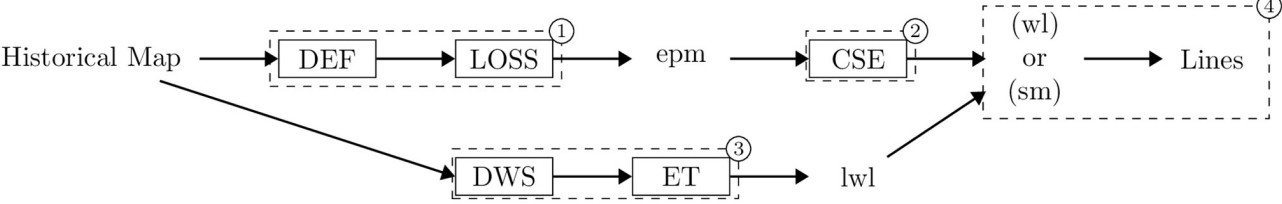

**Fig 1. Our proposed pipeline.** DEF: deep edge filter; LOSS: binary cross entropy loss or topology-oriented losses; DWS: deep watershed; WS: watershed segmentation (Meyer Watershed); ET: edge thinning; LINES: vector output; epm: edge probability map(image of likelihood); lwl: learned watershed levels; (wl) or (sm): watershed lines or saliency maps; The number indicates in the top right of the blocks shows the stages of the processes from 1–4. (Capital letters with box represents the methods and other represents the intermediate results).

## 3.2 Dataset of historical maps

The dataset (DOI:10.5281/zenodo.8325527) used in this paper is taken from paper [5] which is one of the collection of Paris atlases. The performance of the different pipelines under test is assessed using the protocol of the ICDAR 2021 competition on historical map segmentation [57]. In particular, we follow the protocol of Task 1 (*Building blocks detection from historical maps*)but use a different dataset, containing fewer images and for which all closed shapes were annotated—not only building blocks. The dataset contains 2 large map images, extracted either from a series of Paris Atlases ("*Atlas Municipal*") dating from 1898 and 1926 (*Atlas municipal des vingt arrondissements de Paris. 1925. Bibliothèque de l'Hôtel de Ville. Ville de Paris*).

Each map image was manually annotated to create 8,362 polygons in total for each closed shape. Such annotation procedure makes it possible to generate the target Edge Probability Map using the boundaries of the polygons, or to assess the final performance of the vectorization process.

The dataset was split into the subsets summarized in Table 2. The training set is an excerpt from the top of the first sheet of the 1926 edition of the *Atlas Municipal*, while the validation set is built using the lower part of this particular sheet which used as early stop mechanism to prevent over-fitting of the networks. For training and validation, we have selected data from a single map but with varying geographic locations, allowing us to evaluate the performance of our designed pipeline under different geographical contexts. The test set is built using the third sheet of the 1898 edition to test the generalizability of the networks for other historical maps with different scanning conditions. It enables us to assess the adaptability of our pipeline. Ideally, the pipeline that exhibits superior performance on the testing dataset will enhance the digitization quality of other maps in this atlas, ultimately optimizing the overall map digitization process.

## 3.3 Evaluation metrics

We mainly follow the evaluation protocol used by [5, 8], which relies on the COCO Panoptic metric from [58]. Indeed, such metric effectively focuses on the number of shapes which are correctly detected (from the point of view of the ground truth) or predicted (from the point of view of the prediction), leaving the relative size of the shapes as an optional extra indication which may or may not be considered. The COCO Panoptic quality (PQ) term is to measure the quality of intersection of union (IoU) between detected and ground truth instances:

$$PQ = \frac{\sum_{(t_i, p_j) \in TP} IoU(t_i, p_j)}{|TP| + \frac{1}{2}|FP| + \frac{1}{2}|FN|},$$

(1)

where the *FP* means the predicted instances have smaller overlap with the ground truth; and *FN* means the ground truth instances does not pair with any predicted instances. The term *PQ* can also be represented as the product of segmentation quality *SQ* and recognition quality *RQ*

**Table 2. Summary of the training, validation and test sets used in this study.**

| Subset | Atlas Municipal |
|---|---|
| train | 4,500px × 9,000px (3343 inst.) |
| val | 3,000px × 9,000px (2183 inst.) |
| test | 6,000px × 5,500px (2836 inst.) |

where:

$$SQ = \frac{\sum_{(t_i, p_j) \in TP} IoU(t_i, p_j)}{|TP|}, \quad RQ = \frac{|TP|}{|TP| + \frac{1}{2}|FP| + \frac{1}{2}|FN|}. \tag{2}$$

When appropriate, we will show Precision and Recall maps computed using the same IoU values as the COCO Panoptic scores, as introduced by [59]. Precision maps will show, for each predicted shape, the value of the highest possible IoU between this predicted shape and every ground truth shape, using a color scale. Recall maps will conversely show, for each ground truth shape, the value of the highest possible IoU between this expected shape and every predicted shape, using the same color scale. These two qualitative indicators, complemented by a study of the detection quality against shape size, can provide deeper insights about the performance of the segmentation systems we are studying.

## 4 Improved training procedure for deep edge filtering

### 4.1 A better training configuration

[5] noted several limitations in their work. We address here the apparent overfitting of the U-Net model with relatively low performance. We introduce four changes in the original training protocol reported by the authors:

- **Data preprocessing:** instead of centering pixel values by subtracting their mean value, we normalize their values in the [0, 1] range by dividing the original RGB values by 255.

- **Weight initialization:** instead of using a Random Normal initialization, we use a Kaiming initialization [60].

- **Batch size:** instead of the batch size of 1, we use a batch size of 4 (following restricted experiments with sizes of 1, 2, 4 and 6).

- **Pre-trained weights:** instead of the weights provided for the seminal from BDCN network, we now rely on the weights from the PyTorch Image Models ("timm") library from [61] for the VGG backbone.

We do not change the data loading strategy, and use the same code and seeds as the original authors. We also use the same loss formulation; a binary cross-entropy with reweighing of imbalanced classes. The results we report compare both training processes, indicated as **"original"** or **"proposed"**.

We follow the training and selection procedure proposed by the original authors, which unfolds as follows.

- **Training:** For a given Deep Edge Filter *DEF*, using the train set, train the network for $M$ epochs. At each epoch $i$, we obtain $DEF_i$ which generates an Edge Probability Map (EPM).

- **DEF selection:** Using the validation set, compute the corresponding set of Edge Probability Maps $EPM_i$ using $DEF_i$ for $i \in \{0, \ldots, M\}$, then using a naive Closed Shape Extractor $CSE_{naive}$ (described hereafter), select the best Deep Edge Filter $DEF_{best}$ based the topological score

(COCO $PQ$) of the predicted shapes:

$$shapes_i \quad = \quad CSE_{naive}(EPM_i)$$
$$DEF_{best} \quad = \quad argmin_i(PQ(shapes_i))$$

- **CSE parameter tuning:** Then, using the best Deep Edge Filter $DEF_{best}$ as a base, restore or recompute $EPM_{best}$, the set of Edge Probability Maps for the validation set, and grid-search for the best $\theta$ parameters of the Meyer Watershed for Closed Shape Extraction ($CSE_{best}$), over the set of possible parameters $\Omega$:

$$shapes_\theta \quad = \quad CSE_\theta(EPM_{best})$$
$$CSE_{best} \quad = \quad argmin_\theta(PQ(shapes_\theta))$$

- **Global evaluation:** The final evaluation on the test set is performed by combining the best Deep Edge Filter $DEF_{best}$ and the best Closed Shape Extractor $CSE_{best}$ to compute the shapes from test set samples.

We use the same naive CSE as the original authors, i.e, a threshold of the EPM at 0.5 followed by a connected component labelling. However, for a fairer comparison with the watershed CSE, we add an edge-thinning step which allows obtaining thin, 1-pixel-large shape boundaries. We use the same Meyer watershed CSE as the original authors, considering the following values for area filtering with value of 50, 100, 200, 300, 400, 500 number of pixels, and for following values for dynamic filtering with value from 1 to 10 correspond to the discrete value of single pixel with step of 1. The area and dynamic filters are the pre-filtering step for removing non-meaningful local minimum for Meyer watershed. The area is used to merge the regions with size lower than a specific area threshold, while the dynamic refers to the water elevation that is used to merge with other regions. This procedure gives us the opportunity to report the performance (on test set) of two different pipelines:

- **Best Meyer watershed for the CSE stage:** This variant reports the performance of the full pipeline previously described, combining the best Deep Edge Filter $DEF_{best}$ and the best Closed Shape Extractor $CSE_{best}$ to compute the shapes from test set samples.

- **Naive connected component labelling for the CSE stage:** This variant reports the performance $DEF_{best}$, combined with the naive Closed Shape Extractor $CSE_{naive}$. The purpose of reporting this simpler pipeline is to confirm the benefit of using an elaborated CSE stage based on some watershed. Please note that, contrary to [5], we do not optimize the EPM threshold parameter and only report results for a 0.5 threshold value.

Based on this procedure, we report in Tables 3 and 4 results for the following Deep Edge Filters, as in the [5]: U-Net, HED and BDCN. For HED and BDCN, we report whether we used **pre-trained weights** as weight initialization before fine-tuning, or train the network **from scratch**.

These results allow drawing the following conclusions.

- The proposed training procedure significantly improves the performance over the original one, exhibiting superior performance no matter which combination of DEF and CSE is used.

**Table 3. COCO Panoptic scores on validation and test set for the training configuration study, using a naive connected component labelling for CSE.** The training configuration from [5] is indicated as "Original", while our proposed method is indicated as "Proposed". The following parameters are static, and their respective columns are hidden: the CSE used is a naive connected component labelling ([5] used a grid search to find the best threshold $\theta$ for EPM binarization while we use a fixed value of 0.5), the loss function is the binary cross entropy, the best DEF is selected using the protocol of [5], no augmentation is performed. For the architectures, * indicates pre-trained variants.

| DEF Archi. | Training config. | CSE Param. | Evaluation Val. set | | | Test set | | |
|---|---|---|---|---|---|---|---|---|
| | | $\theta$ | PQ | SQ | RQ | PQ | SQ | RQ |
| U-Net | Original | 0.9 | 34.8 | 80.5 | 43.3 | 8.1 | 78.2 | 10.4 |
| HED | Original | 0.3 | 23.2 | 76.5 | 30.3 | 14.0 | 74.8 | 18.8 |
| HED* | Original | 0.7 | 27.6 | 77.6 | 35.6 | 16.2 | 76.1 | 21.3 |
| BDCN | Original | 0.9 | 27.7 | 80.6 | 34.4 | 10.2 | 80.4 | 12.7 |
| BDCN* | Original | 0.8 | 27.6 | 82.1 | 33.7 | 8.9 | 82.8 | 10.7 |
| U-Net | Proposed | 0.5 | 46.8 | 87.5 | 53.5 | 41.2 | 85.4 | 48.2 |
| HED | Proposed | 0.5 | 52.2 | 86.8 | 60.2 | 42.7 | 85.2 | 50.1 |
| HED* | Proposed | 0.5 | 32.4 | 87.0 | 37.3 | 44.5 | 85.2 | 52.3 |
| BDCN | Proposed | 0.5 | 51.4 | 86.5 | 59.5 | 43.4 | 85.2 | 50.9 |
| BDCN* | Proposed | 0.5 | 55.7 | 87.0 | 64.0 | 41.4 | 86.1 | 48.1 |

- With the proposed training procedure, the HED architecture is more accurate, and the naive Connected Component labelling exhibits the best performance for the CSE stage.

- For both HED and BDCN architectures, we confirm the superiority of the pre-trained networks which always exhibit a better performance than when trained from scratch.

- The U-Net architecture exhibits much higher generalization performance, and now achieves the best overall performance on the test set with a COCO PQ score of 46.7%.

## 4.2 Joint optimization of deep edge filtering and closed shape extraction

As previously mentioned, [5, 8] optimized each stage of their segmentation pipeline independently. To enable the joint optimization of both stages, we propose to run a parameter selection (using a grid search) for the CSE stage, for each epoch of the training of each Deep Edge

**Table 4. COCO Panoptic scores on validation and test set for the training configuration study, using the Meyer Watershed (MWS) for CSE.** The training configuration from [5] is indicated as "Original" while our proposed method is indicated as "Proposed". The following parameters are static, and their respective columns are hidden: the loss function is the binary cross entropy, the best DEF is selected using the protocol of [5], no augmentation is performed. For the architectures, * indicates pre-trained variants.

| DEF Archi. | Training config. | CSE Param. | | Evaluation Val. set | | | Test set | | |
|---|---|---|---|---|---|---|---|---|---|
| | | $\sigma$ | $\delta$ | PQ | SQ | RQ | PQ | SQ | RQ |
| U-Net | Proposed | 50.0 | 8.0 | 59.8 | 87.7 | 68.2 | 46.7 | 86.9 | 53.7 |
| HED | Proposed | 400.0 | 10.0 | 47.5 | 86.8 | 54.7 | 41.0 | 85.2 | 48.1 |
| HED* | Proposed | 400.0 | 10.0 | 51.5 | 87.5 | 58.9 | 43.9 | 86.2 | 50.9 |
| BDCN | Proposed | 400.0 | 10.0 | 48.9 | 86.9 | 56.3 | 41.3 | 85.8 | 48.1 |
| BDCN* | Proposed | 400.0 | 10.0 | 54.7 | 88.6 | 61.7 | 46.4 | 87.1 | 53.3 |
| U-Net | Original | 50.0 | 0.0 | 56.6 | 87.7 | 64.5 | 18.3 | 85.2 | 21.4 |
| HED | Original | 200.0 | 10.0 | 50.5 | 87.2 | 57.9 | 35.6 | 84.6 | 42.0 |
| HED* | Original | 300.0 | 8.0 | 52.8 | 87.6 | 60.3 | 38.4 | 85.5 | 44.9 |
| BDCN | Original | 300.0 | 8.0 | 52.5 | 87.8 | 59.8 | 34.9 | 85.8 | 40.6 |
| BDCN* | Original | 300.0 | 7.0 | 53.0 | 88.1 | 60.1 | 37.8 | 86.4 | 43.8 |

Filter. This process is implemented by merging steps 2 and 3 from the original training procedure given in Section 4. At the end of each epoch $i$ of the training of the DEF, generate the set of Edge Probability Maps $EPM_i$ for the validation set using $DEF_i$, then, using a grid-search, select the parameters for the CSE stage which leads to the best performance (based on the COCO Panoptic score) on the validation set. The overall, final performance is reported on the test set (which was never used in any part of the joint optimization).

$$EPM_i = DEF_i(\text{val. set})$$

$$shapes_{i,\theta} = CSE_\theta(EPM_i)$$

$$\{DEF, CSE\}_{best} = argmin_{i,\theta}(PQ(shapes_{i,\theta}))$$

We compare the following segmentation pipelines:

- **Baseline** (independent optimization of DEF and CSE stages). The best DEF parameters (epoch) are selected based on the COCO Panoptic score obtained on the validation set, using a simplified CSE stage: thresholding the EPM with a fixed value of 0.5, then extracting the shapes using CC labelling. The best CSE parameters (for the watershed extractor) are then computed for one DEF model only. This corresponds to the **Best Meyer Watershed ("best MWS")** variant in Section 4.

- **Joint Optimization** This variant tests all possible combinations of DEF and CSE configurations for each epoch, reaching the best possible combination of DEF and CSE systems.

For the DEF stage, the set of possible parameters is defined as the different model trainings obtained at each epoch, and for the CSE stage, we consider, as [5, 8], area filtering and dynamic filtering for the watershed stage. Area filtering consists in discarding shapes with an area smaller than a particular value (expressed in pixels here), and dynamic filtering discards the shapes whose dynamic is smaller than a particular value (The dynamic value is in between [0, 255]). We used 6 different values for area filtering ranging from 50 pixels to 500 pixels (50, 100, 200, 300, 400, 500) (corresponding to small shapes according the dataset) and 10 values for dynamic ranging from 1 to 10 with step of 1, exactly as in [5], for the Meyer watershed. All DEF networks are training using a binary cross-entropy loss.

Quantitative and qualitative results are reported in Table 5 and Fig 2, for the same U-Net, HED, and BDCN networks as previously, and show that the systematic superiority of the joint optimization strategy over the baseline approach on the validation set (enforced by the selection protocol) does not always guarantee to reach the best performance on the test set for the HED network and the BDCN network trained from scratch. However, it enables to further

**Table 5. COCO Panoptic scores on validation and test set for the joint optimization study, using the Meyer Watershed for CSE and Joint Optimization (JO) for DEF selection.** The following parameters are static, and their respective columns are hidden: we use our proposed training configuration, the loss function is the binary cross entropy, no augmentation is performed. For the architectures, * indicates pre-trained variants.

| DEF Archi. | CSE Param. | | Evaluation Val. set | | | Test set | | |
|---|---|---|---|---|---|---|---|---|
| | $\sigma$ | $\delta$ | PQ | SQ | RQ | PQ | SQ | RQ |
| U-Net | 50.0 | 10.0 | 60.4 | 88.2 | 68.5 | 47.1 | 86.8 | 54.3 |
| HED | 400.0 | 10.0 | 47.6 | 86.8 | 54.9 | 40.8 | 85.0 | 47.9 |
| HED* | 400.0 | 10.0 | 51.8 | 87.5 | 59.2 | 43.7 | 86.2 | 50.7 |
| BDCN | 400.0 | 10.0 | 49.1 | 86.9 | 56.5 | 41.1 | 86.0 | 47.8 |
| BDCN* | 400.0 | 9.0 | 55.0 | 88.5 | 62.1 | 47.0 | 87.3 | 53.8 |

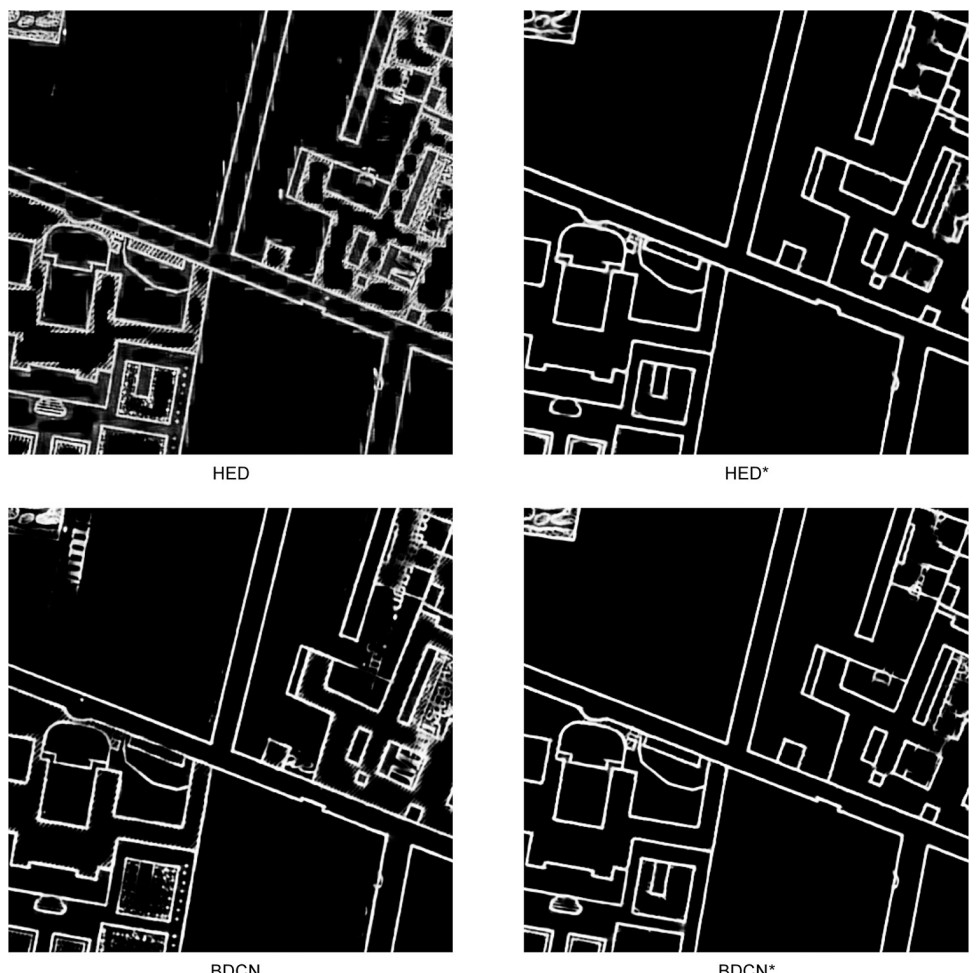

**Fig 2. Predicted EPMs (Scale: 1:5,000, size: 500 px × 500 px) with U-Net, HED and BDCN with (\*) or without pre-trained weights.** Pre-trained model produces less noise on the edges.

push the COCO PQ score of both U-Net and BDCN architectures, reaching a value of 47.1% and 47.0% respectively, and defines a new reference score with 0.4 percentage point of improvement over the best result.

## 5 Alternate architectures and training for deep edge filtering

### 5.1 Vision transformers

Vision Transformers tend to exceed CNN architectures in many tasks, thanks to their multi-head self-attention with images [41]. We compare here two recent Transformer architecture against the best U-Net variant obtained in Section 4.2: the Vision Image Transformer (ViT), and the Pyramid Vision Transformer (PVT) [43] that can be easily adapted dense prediction task compared to ViT.

As training Transformer architectures from scratch requires large amount of training data, we initialize our network using weights pre-trained on the Cityscapes dataset [62] to speed up the convergence of the models. We use the ADAM optimizer for both Transformers.

**Table 6. COCO Panoptic scores on validation and test set for transformer architectures.** The following parameters are static, and their respective columns are hidden: we use the Meyer Watershed (MWS) for CSE and Joint Optimization (JO) for DEF selection, we use our proposed training configuration, the loss function is the binary cross entropy, no augmentation is performed. For the architectures, * indicates pre-trained variants.

| DEF Archi. | CSE Param. | | Evaluation Val. set | | | Test set | | |
|---|---|---|---|---|---|---|---|---|
| | $\sigma$ | $\delta$ | PQ | SQ | RQ | PQ | SQ | RQ |
| U-Net | 50.0 | 10.0 | 60.4 | 88.2 | 68.5 | 47.1 | 86.8 | 54.3 |
| ViT* | 500.0 | 10.0 | 38.6 | 80.9 | 47.8 | 34.7 | 80.4 | 43.1 |
| PVT* | 400.0 | 9.0 | 45.7 | 85.4 | 53.5 | 36.6 | 83.0 | 44.2 |

Quantitative and qualitative results, summarized in Table 6 and Fig 3, show that despite their larger receptive field, transformer architectures reach much lower validation and test scores in our experiments, compared to the traditional, U-Net architecture. This low performance is caused by the fact that ViT has a low-resolution output, and that ViT and PVT may require a much larger dataset for fine-tuning. However, the performance of these architectures

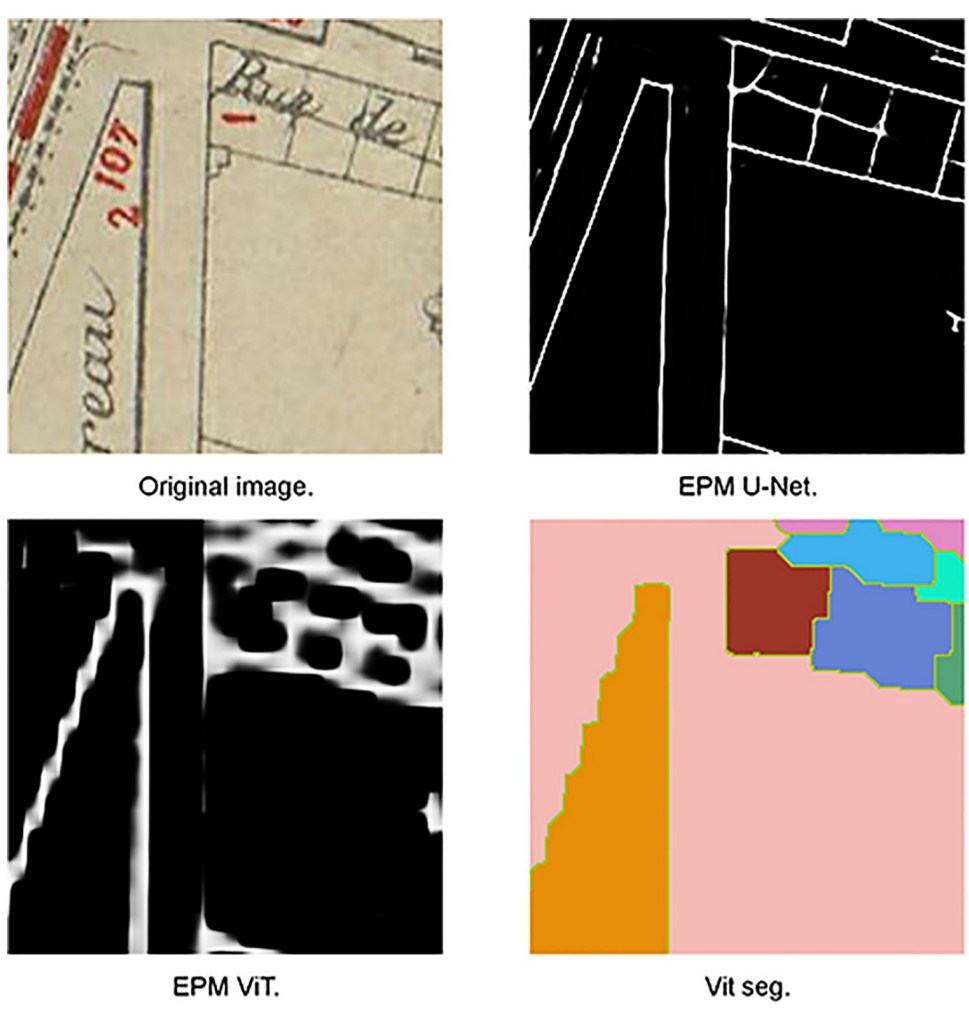

Original image.

EPM U-Net.

EPM ViT.

Vit seg.

**Fig 3. Zigzag effect of EPM produced by ViT architecture.**

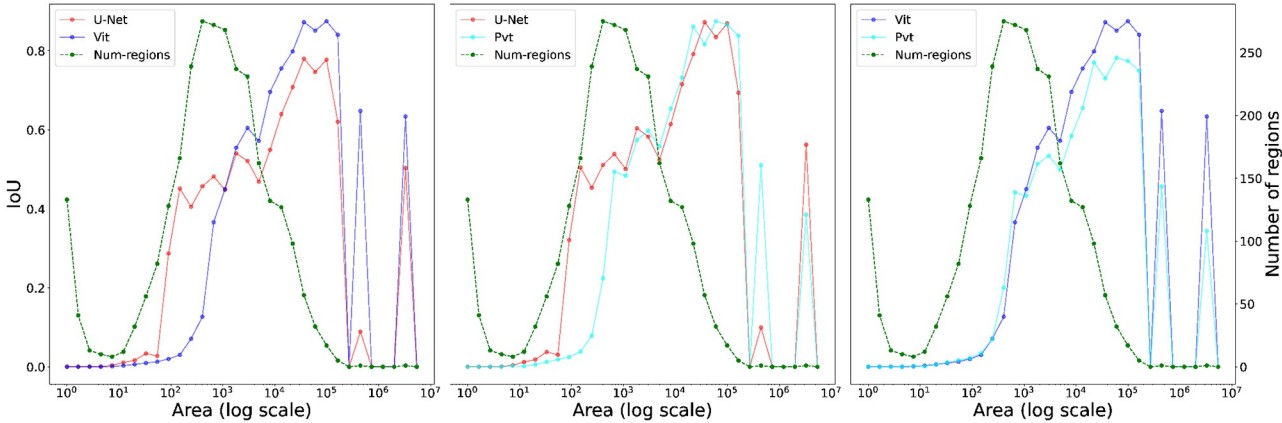

**Fig 4. Shape statistics of convolutional-based (U-Net) and transformer-based (ViT and PVT) models.** The figure represents the average IoU with step of 0.5. The green dash line corresponds to the distribution of number of regions with different area values. ViT and PvT transformer-based architecture have better performance for detecting large objects compare to U-Net architecture where U-Net are good at detecting small objects (compare to transformer).

is not always worse than the U-Net one, especially for larger shapes (log(area) > 15), where ViT outperforms U-Net. Some combination of these systems may be possible to obtain the best possible performance, keeping only smaller objects from U-Net and larger ones from ViT (shown in Fig 4). Regarding PVT, its overall performance is not better compared to the conditions (baseline U-Net) we tested it against. Looking at the values of the best parameters for the area and dynamic filtering, we can see that both ViT and PVT models require stronger filtering compared to U-Net. This suggests that Transformer-based models are suffering from an important amount of noise image in the predicted EPMs.

## 5.2 Topology-preserving loss functions

Several loss functions were designed in the literature [38, 39, 49] to better comply with the topological requirements of several tasks. Most of these loss functions need to be combined with a classical binary cross-entropy loss. We use a λ parameter to scale the value of the topological loss in the final loss.

$$\mathcal{L} = \mathcal{L}_{\mathrm{BCE}} + \lambda \mathcal{L}_{\mathrm{topological}}.$$

We report here the performance of a U-Net network trained alternatively with each of the following loss functions: (a)**MOSIN**: We set the λ = 0.001; (b) **BALoss**: We set the λ = 1000; (c) **Topoloss**: We set the λ = 0.01; (d) **Binary cross-entropy**: the baseline. All variants are trained using ADAM optimizer except for the Topoloss one, which is trained using SGD according to authors' recommendation. We set the initial learning rate to $1 \cdot 10^{-4}$, a momentum of 0.9 and a weight decay of $1 \cdot 10^{-5}$.

We also report results for the two following pipelines:

- **Naive connected component labelling (Abbr. cc) for the CSE stage.** This variant, as detailed in Section 4 improved training for DEF, enables to measure the quality of the EPM, from a topological point of view.

- **Joint Optimization.** This variant, detailed in Section 4.2, is based on the joint optimization of the Deep Edge Filter and Closed Shape Extraction stages, and leads to the best performance.

**Table 7. COCO Panoptic scores on validation and test set for study on topological loss functions.** The following parameters are static, and their respective columns are hidden: we use the Meyer Watershed (MWS) for CSE and Joint Optimization (JO) for DEF selection, we use our proposed training configuration, no augmentation is performed. For the architectures, * indicates pre-trained variants: the network is trained first using binary cross-entropy, then using a custom loss.

| DEF Archi. | Loss function | CSE Param. | | Evaluation Val. set | | | Test set | | |
|---|---|---|---|---|---|---|---|---|---|
| | | $\sigma$ | $\delta$ | PQ | SQ | RQ | PQ | SQ | RQ |
| U-Net | binxent | 50.0 | 10.0 | 60.4 | 88.2 | 68.5 | 47.1 | 86.8 | 54.3 |
| U-Net* | bal | 50.0 | 1.0 | 63.1 | 87.6 | 72.0 | 45.6 | 86.3 | 52.9 |
| U-Net* | topo | 100.0 | 6.0 | 59.9 | 88.1 | 68.0 | 36.9 | 84.2 | 43.8 |
| U-Net* | mosin | 50.0 | 1.0 | 57.7 | 88.3 | 65.3 | 36.0 | 87.4 | 41.2 |

Table 7 summarizes the results for the different variants. It shows that despite the BALoss variant is able to achieve a better performance on the validation set, it remains slightly less performant than the baseline version on the test set. Hence, the traditional binary cross-entropy seems to remain the more appropriate loss function for this training protocol, even if the BALoss seems promising for a dataset which does not have much domain shift.

## 5.3 Data augmentations strategies

Online data augmentation at training time can improve the generalization performance of the Deep Edge Filters. We consider the following geometric augmentation techniques, and the examples of augmented figures are shown in Fig 5: affine transformation, homography transformation (full perspective), and thin-plate splines (TPS) transformation. We also consider contrast change for signal augmentation. Likewise, we study the effect of separate and combined geometric and contrast augmentation. The joint optimization protocol with U-Net model is described in Section 4.2.

Furthermore, we choose contrast $g(x) = \alpha \cdot f(x)$ through image $f(x)$ range of $\alpha$ from 0.8 to 1.2 (to prevent unrealistic dark or bright cases which will not occur in scanned maps) with uniform distribution. For the setting of geometric transformation, we change settings in the work of [54] to prevent large geometric transform which is not happened in different historical map sheets. For affine transformation, we choose rotation angle $\theta \sim \mathcal{U}(-10°, 10°)$ ($\mathcal{U}$ is the uniform distribution.), translation $t_x, t_y \sim \mathcal{U}(-0.1, 0.1)$, anisotropic scaling factor $\lambda_1, \lambda_2 \sim \mathcal{U}(0.9, 1.1)$ and shear angle $\phi \sim \mathcal{U}(-10°, 10°)$. For homography transformation, we add a random translation $\sigma_x, \sigma_y \sim \mathcal{U}(-0.1, 0.1)$ to four control points in the corner. For TPS transformation, we use 9 points with random translation $\delta_x, \delta_y \sim \mathcal{U}(-0.1, 0.1)$ to prevent strong deformations.

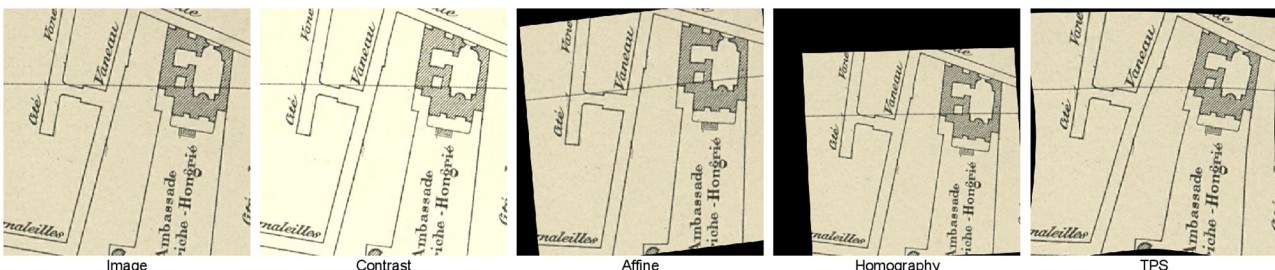

**Fig 5. Image examples with four different augmentation methods.**

**Table 8. COCO Panoptic scores on validation and test set for the augmentation study.** The following parameters are static, and their respective columns are hidden: model architecture is U-Net (trained from scratch), we use the improved training variant, the loss function is the binary cross entropy, the best DEF is selected using joint optimization, and Meyer Watershed (MWS) is used for CSE.

| DEF Augmentation Contrast streching | Geometric transform | CSE Param. | | Evaluation Val. set | | | Test set | | |
|---|---|---|---|---|---|---|---|---|---|
| | | $\sigma$ | $\delta$ | PQ | SQ | RQ | PQ | SQ | RQ |
| no | none | 50.0 | 10.0 | 60.4 | 88.2 | 68.5 | 47.1 | 86.8 | 54.3 |
| yes | none | 100.0 | 6.0 | 57.3 | 88.2 | 65.0 | 47.2 | 86.7 | 54.4 |
| no | Aff. | 100.0 | 9.0 | 61.0 | 87.9 | 69.4 | 47.7 | 86.5 | 55.1 |
| yes | Aff. | 100.0 | 10.0 | 61.1 | 88.1 | 69.4 | 50.7 | 86.8 | 58.5 |
| no | Hom. | 200.0 | 10.0 | 58.4 | 87.9 | 66.5 | 49.6 | 86.9 | 57.1 |
| yes | Hom. | 200.0 | 10.0 | 59.5 | 88.2 | 67.4 | 50.4 | 86.7 | 58.2 |
| no | TPS | 100.0 | 10.0 | 59.8 | 88.3 | 67.8 | 47.9 | 86.9 | 55.1 |
| yes | TPS | 100.0 | 7.0 | 59.6 | 88.2 | 67.5 | 51.1 | 86.8 | 58.8 |

Table 8 reports the results for the various combinations. All methods lead to improved performance on the test set. The combined use of contrast and TPS augmentations leads to the best COCO PQ score, reaching 51.1%.

## 5.4 Lightweight architecture

The superior performance of the U-Net architecture is vital for Deep Edge Filtering stage. We also test mini-U-Net with a lower amount of parameters of this architecture. We report the result of an additional experiment with a smaller variant of the U-Net architecture, referred to as **"mini U-Net"**. This **"mini U-Net"** contains only 4 M parameters, compared to the 17 M parameters of the baseline. We train both architectures with the joint optimization protocol described in Section 4.2.

Results, summarized in Table 9, show that despite a much smaller footprint, the smaller implementation is still very competitive, with a drop of 2 points, from 47.1 to 45.1, of the COCO PQ score on the test set. As a conclusion, with some applications which require the limited memory, mini U-Net would be a good choice.

## 6 Alternate watershed implementations for closed shape extraction

[5, 8] considered two approaches for CSE: a naive, baseline system based on connection component labeling (after thresholding the results of the DEF stage). A more elaborate one based on the Meyer watershed algorithm, which can recover weak edges as long as they are stronger than a minimal dynamic and enable to create shapes larger than a minimal area. We consider one extra approach: a **Deep Watershed Transform**, which integrates the DEF and CSE stages in a single pipeline.

**Table 9. COCO Panoptic scores on validation and test set for U-Net variants.** The following parameters are static, and their respective columns are hidden: we use our proposed training configuration, the loss function is the binary cross entropy, no augmentation is performed, DEF selection is performed with Joint Optimization (JO), and we use the Meyer Watershed (MWS) for CSE.

| DEF Archi. | CSE Param. | | Evaluation Val. set | | | Test set | | |
|---|---|---|---|---|---|---|---|---|
| | $\sigma$ | $\delta$ | PQ | SQ | RQ | PQ | SQ | RQ |
| U-Net | 50.0 | 10.0 | 60.4 | 88.2 | 68.5 | 47.1 | 86.8 | 54.3 |
| mini U-Net | 100.0 | 10.0 | 56.7 | 87.7 | 64.6 | 45.1 | 86.0 | 52.5 |

**Table 10. COCO Panoptic scores on validation and test set for U-Net+Meyer Watershed vs Deep Watershed.** We use the MWS as a post-processing without filtering on Deep Watershed outputs to thin the prediction edges. The following parameters are static, and their respective columns are hidden: we use our proposed training configuration, the loss function is the binary cross entropy, no augmentation is performed, and DEF selection is performed with Joint Optimization (JO).

| DEF Archi. | CSE Method | Param. | | Evaluation Val. set | | | Test set | | |
|---|---|---|---|---|---|---|---|---|---|
| | | $\sigma$ | $\delta$ | PQ | SQ | RQ | PQ | SQ | RQ |
| U-Net | MWS | 50.0 | 10.0 | 60.4 | 88.2 | 68.5 | 47.1 | 86.8 | 54.3 |
| DWS | MWS | 0.0 | 0.0 | 54.0 | 87.4 | 61.7 | 28.5 | 84.9 | 33.5 |

The training strategy from [56] is to train a direction network and a watershed network separately, then perform fine-tuning in an end-to-end style. However, we found that the overfitting of both direction and watershed networks make fine-tuning hard to converge, while training the direction network first and fine-tuning the two networks together can achieve better performance in terms of COCO Panoptic score. Furthermore, instead of using the original architecture from [56], we used two U-Net architectures for both direction and watershed networks to maintain better spatial information for each pixel. We trained the direction network using an ADAM optimizer with the initial learning rate of $1 \cdot 10^{-5}$, a momentum of 0.9 and a weight decay of $1 \cdot 10^{-5}$. Then, end-to-end fine-tuning used the same settings, except for a smaller learning rate of $1 \cdot 10^{-6}$. In order to ensure a fair comparison with other approaches, we generated object boundaries with the following process: we first performed the equivalent of a "watershed cut" by selecting the highest value on the learned watershed levels (this creates thick boundaries), then we performed an edge-thinning to recover thin, 1-pixel large object boundaries.

We compare in Table 10 the results of the Deep Watershed approach and of the leading approach, composed of a U-Net combined with a Meyer Watershed, trained using the joint optimization. Despite encouraging performance, the Deep Watershed fails to generalize on our test set, reaching much lower performance than our leading approach. It is due to the fact that deep watershed learns an approximation function which transforms image into watershed level but without providing any topology guarantee (closed shapes) in the final prediction of watershed levels due to limited spatial context.

## 7 Conclusion

In this paper, we comprehensively evaluated the most suitable solution for automatic and efficient vectorization of historical maps. We explored the numerous possibilities on both sides of the two-stage framework proposed in [5]. We tested the main state-of-the-art variants either separately or simultaneously. We found large improvements for variants **joint optimization (Deep Edge Filtering + Meyer watershed)**, minor improvements for variants of **topology-related loss functions**, and decrease in performances for variants **Transformer-based architecture, lighter version of U-Net and deep watershed segmentation techniques.Joint optimization** is proved to be an effective tool to select the best combinations of Edge Probability Maps and watershed segmentation. Moreover, the **BALoss** is the best loss among all the topology-related variants which activates the correct boundaries in the predicted EPM and decreases the number of weak pixels that leads to topology failures. Also, the combination of topology-related loss functions with the joint optimization process provides an improvement. We tested **Transformer-based architectures** with self-attention mechanism, and they are proved to be effective in detecting large instances compared to Convolutional Neural Networks. However, they fail in detecting fine-grained local cues which leads to skip finding objects with small areas. Concerning **data augmentation** strategies, contrast stretch combines with thin-plate-spline has the most generalizability. A lighter version of U-Net (**mini-U-Net**)

was tested, and it showed similar performance compared to the original version of U-Net, which can be useful in some historical map applications with limited memory. Nonetheless, due to the complexity and challenges structure of historical maps, more parameters of networks are necessary to receive better filtering performance. We also modified the end-to-end learnable watershed level (**Deep watershed**) for adapting to the historical map vectorization tasks. We shown that worse performance is achieved compared to our joint optimization process since it does not guarantee the topology properties in the predicted likelihood image. We wish this work can help researchers to extract high vectorization quality from historical maps as well as leverage the human annotation effort.

Digitized maps provide numerous research opportunities that offer substantial benefits to different sectors. These opportunities include analyzing historical maps to uncover the growth and transformation of cities and urban areas, which can be particularly valuable for **municipal planning**. They are also a valuable resource for **archaeologists**, aiding in the mapping and preservation of archaeological sites and cultural heritage locations. Furthermore, digitized maps facilitate the examination of historical maps, enabling a deeper understanding of past geopolitical boundaries and territories. This capability is beneficial for **historians** and **cartographers** alike as they explore the history of cartography and mapmaking processes. To conclude, it is important to note that these applications only scratch the surface of the wide-ranging possibilities made accessible through digitized maps. The automated map digitization process will greatly accelerate research progress across numerous fields.

The data, codes, and instructions that support the findings of this study are available at the public Zenodo platform https://zenodo.org/records/10567765 and Github repository https://github.com/soduco/Benchmark_historical_map_vectorization.

## Supporting information

**S1 Appendix. Deep edge filters and training parameters.** Full list of experimental results. (PDF)

## Author Contributions

**Conceptualization:** Yizi Chen, Joseph Chazalon, Edwin Carlinet, Minh Ôn Vũ Ngoc, Clément Mallet, Julien Perret.

**Data curation:** Yizi Chen, Joseph Chazalon, Edwin Carlinet, Clément Mallet, Julien Perret.

**Formal analysis:** Yizi Chen, Joseph Chazalon, Edwin Carlinet, Minh Ôn Vũ Ngoc, Clément Mallet, Julien Perret.

**Funding acquisition:** Julien Perret.

**Investigation:** Yizi Chen, Joseph Chazalon, Edwin Carlinet, Minh Ôn Vũ Ngoc, Clément Mallet, Julien Perret.

**Methodology:** Yizi Chen, Joseph Chazalon, Edwin Carlinet, Minh Ôn Vũ Ngoc, Clément Mallet, Julien Perret.

**Project administration:** Joseph Chazalon, Clément Mallet, Julien Perret.

**Resources:** Joseph Chazalon, Edwin Carlinet, Clément Mallet, Julien Perret.

**Software:** Yizi Chen, Joseph Chazalon, Edwin Carlinet, Minh Ôn Vũ Ngoc, Clément Mallet, Julien Perret.

**Supervision:** Joseph Chazalon, Clément Mallet, Julien Perret.

**Validation:** Yizi Chen, Joseph Chazalon, Edwin Carlinet, Minh Ôn Vũ Ngoc, Clément Mallet, Julien Perret.

**Visualization:** Yizi Chen, Joseph Chazalon, Julien Perret.

**Writing – original draft:** Yizi Chen, Joseph Chazalon, Clément Mallet, Julien Perret.

**Writing – review & editing:** Yizi Chen, Joseph Chazalon, Clément Mallet, Julien Perret.

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
