## [Decision Letter · Decision Letter 0]

30 Aug 2023

PONE-D-23-22523Automatic vectorization of historical maps: a benchmarkPLOS ONE

Dear Dr. CHEN,

Thank you for submitting your manuscript to PLOS ONE. After careful consideration, we feel that it has merit but does not fully meet PLOS ONE’s publication criteria as it currently stands. Therefore, we invite you to submit a revised version of the manuscript that addresses the points raised during the review process.

We look forward to receiving your revised manuscript.

Kind regards,

Yawen Lu, Ph.D

Academic Editor

PLOS ONE

Journal Requirements:

   "This work was supported by the French National Research Agency (ANR), as part of the SoDUCo project (grant ANR-18-CE38-0013)."

4. Please ensure that you refer to Figures 4-7 in your text as, if accepted, production will need this reference to link the reader to the figure.

5. We note that Figures 1 and 3 in your submission contain [map/satellite] images which may be copyrighted. All PLOS content is published under the Creative Commons Attribution License (CC BY 4.0), which means that the manuscript, images, and Supporting Information files will be freely available online, and any third party is permitted to access, download, copy, distribute, and use these materials in any way, even commercially, with proper attribution. For these reasons, we cannot publish previously copyrighted maps or satellite images created using proprietary data, such as Google software (Google Maps, Street View, and Earth). For more information, see our copyright guidelines: http://journals.plos.org/plosone/s/licenses-and-copyright.

a. You may seek permission from the original copyright holder of Figures 1 and 3 to publish the content specifically under the CC BY 4.0 license.  

Additional Editor Comments:

There is some disagreement between two reviewers. One reviewer votes to reject due to inappropriate scope. Another reviewer raises several concerns, such as the discussion of the specific maps, the discussion of the research applications of this work, and some writing issues. In addition, I found the image quality in the appendix to be very poor and blurred. Overall, I think the paper is suitable and in good quality. I expect the authors to address such concerns in order to have a quick turnaround.

Reviewers' comments:

Reviewer's Responses to Questions

**Comments to the Author**

1. Is the manuscript technically sound, and do the data support the conclusions?

Reviewer #1: Yes

Reviewer #2: Yes

2. Has the statistical analysis been performed appropriately and rigorously? 

Reviewer #1: N/A

Reviewer #2: Yes

3. Have the authors made all data underlying the findings in their manuscript fully available?

Reviewer #1: Yes

Reviewer #2: Yes

4. Is the manuscript presented in an intelligible fashion and written in standard English?

Reviewer #1: Yes

Reviewer #2: Yes

5. Review Comments to the Author

Reviewer #1: Dear authors,

Your manuscript is wonderful and your findings are clear with what you all have done. However, in my opinion, it does not suit to the journal. You may try any relevant journals which will help to make your work recognized better.

Reviewer #2: Thank you to the authors for submitting this paper. It's well organized and presents innovating research on historic map vectorization. The authors provides useful background on map digitization and successfully situate their work in the context of prior research.

Overall, it's a very complete paper, and the model construction and testing process is outlined in great detail. It's a great fit for the journal and I think it's in good shape for publication.

I'd recommend a couple revisions. Some discussion of the specific maps that form the train, val, and test data would be useful, partly in order to specify the scope conditions for the digitization model. Where else do you expect the model to perform - maps of the same publisher, scale, or vintage, in particular? All historic building-level maps of a similar scale?

Relatedly, a short discussion about the research applications of the authors' work would improve the conclusion section. The authors comprehensively demonstrate the effectiveness of their approach; the conclusion would be stronger by outlining clearly who will benefit from it. (We can easily guess where it may be applied, but outlining this at the end of the paper will make it a stronger argument, in my opinion).

See below for some line edits, mostly related to sentence syntax. Another light round of typo/language/syntax editing would make the paper read more clearly in a few places.

Line edits

Abstract:

"highly informative information" doesn't really work as a phrase.

"opens numerous ways" is also a strange way to phrase it.... Maybe "opens numerous new approaches" ?

"that reliably extract closed shaped"...... "shaped" should be "shapes"

"loss functions, among which vision transformers"...... How about "including vision transformers" ?

Main body of paper:

Lines 98-101: breaking this sentence into two - or adding several commas - will make it much easier to digest.

6. PLOS authors have the option to publish the peer review history of their article (what does this mean?). If published, this will include your full peer review and any attached files.

Reviewer #1: **Yes: **Shaban Ahmad

Reviewer #2: No

---

## [Author Response · Author response to Decision Letter 0]

23 Nov 2023

Dear academic editor and reviewer(s),

First, we extend our appreciation to the academic editor for concurring that this work aligns with the criteria for publication in the PLOS ONE journal. 

Additionally, we wish to express our sincere gratitude for the invaluable feedback offered by both the academic editor and the reviewers. 

This response letter comprehensively addresses all the concerns and points raised by the academic editor and the reviewer(s).

To editor:

1. Renaming the files to align with PLOS ONE's style guidelines. We have created new .tif images, which are included as attachments with this email. We would like to highlight an issue concerning our manuscript. Specifically, we have noticed that our figures appear blurred in the built version but exhibit satisfactory quality when downloaded. We are seeking guidance on whether this presents a problem and, if so, how best to address it.

2. Stating financial disclosure: This work was supported by the French National Research Agency (ANR), as part of the SoDUCo project (grant ANR-18-CE38-0013). The funders had no role in study design, data collection and analysis, decision to publish, or preparation of the manuscript.

3. We added DOI number of the dataset: 10.5281/zenodo.8325527; We have also included the repository in the abstract, enabling quick access to all of our code and pre-trained models for replicating the results presented in this paper.

4. Figures 4 and 7 have been referenced in the article.

5. The original image is sourced from BHVP(Atlas municipal des vingt arrondissements de Paris. 1925. Bibliothèque de l’Hôtel de Ville. Ville de Paris), and we have been explicitly granted permission to use it (we have included the approval document with the filename `Permission\\_of\\_using\\_atlas\\_municipal.pdf'.).

As we are solely utilizing a cropped portion of the map image, we do not face any copyright issues of figures 1 and 3 in relation to the original atlas map image from Atlas Municipal.

In response to the editor's recommendations, we incorporated the following sentence into the figure caption: "This figure is cropped from the Atlas Municipal for illustrative purposes only".

Please let us know if you find that this solution does not adequately address your concerns.

6. Thank you for bringing up this matter. 

I want to clarify that we have not referenced any retracted papers in this article.

To reviewers 1:

1. Dear authors, Your manuscript is wonderful and your findings are clear with what you all have done. However, in my opinion, it does not suit to the journal. You may try any relevant journals which will help to make your work recognized better.

A: We appreciate your concerns and the invaluable feedback you've provided. 

We firmly believe that our work harmoniously aligns with the scope and mission of PLOS ONE, a recognition shared by the academic editor and other reviewer.

To reviewers 2: 

1. I'd recommend a couple revisions. Some discussion of the specific maps that form the train, val, and test data would be useful, partly in order to specify the scope conditions for the digitization model.

A: The scope condition in this article is tailored to a specific historical map atlas named Atlas Municipal with scale 1:5000 that characterized by minimal color and texture information, with objects primarily delineated by distinct boundaries. Given the extensive volume of maps that require digitization, our ultimate objective is to develop a versatile pipeline capable of delivering high-quality digitization results across this map collection. To achieve this goal, we have curated a dataset comprising training, validation, and testing data to assess the digitization quality of various pipeline variants. We have added this extra discussion into the article. 

"For training and validation, we have selected data from a single map but with varying geographic locations, allowing us to evaluate the performance of our designed pipeline under different geographical contexts. For the testing dataset, we have chosen maps from different years and geographic locations. It enables us to assess the adaptability of our pipeline. Ideally, the pipeline that exhibits superior performance on the testing dataset will enhance the digitization quality of other maps in this atlas, ultimately optimizing the overall map digitization process".

Where else do you expect the model to perform - maps of the same publisher, scale, or vintage, in particular? All historic building-level maps of a similar scale?

We have only tested this pipeline in the datasets created by Atlas Municipal. 

Indeed, expanding the scope of our research, we plan to explore additional datasets (with different publisher, scale, or vintage maps) using the same pipeline as part of our future investigations.

2. Relatedly, a short discussion about the research applications of the authors' work would improve the conclusion section. The authors comprehensively demonstrate the effectiveness of their approach; the conclusion would be stronger by outlining clearly who will benefit from it. (We can easily guess where it may be applied, but outlining this at the end of the paper will make it a stronger argument, in my opinion).

A: We sincerely appreciate your feedback. In response, we have incorporated the following sentence into the conclusion section, enhancing its overall strength compared to the previous version.

"Digitized maps provide numerous research opportunities that offer substantial benefits to different sectors. These opportunities include analyzing historical maps to uncover the growth and transformation of cities and urban areas, which can be particularly valuable for municipal planning. They are also a valuable resource for archaeologists, aiding in the mapping and preservation of archaeological sites and cultural heritage locations. Furthermore, digitized maps facilitate the examination of historical maps, enabling a deeper understanding of past geopolitical boundaries and territories. This capability is beneficial for historians and cartographers alike as they explore the history of cartography and mapmaking processes. To conclude, it is important to note that these applications only scratch the surface of the wide-ranging possibilities made accessible through digitized maps. The automated map digitization process will greatly accelerate research progress across numerous fields."

3. See below for some line edits, mostly related to sentence syntax. Another light round of typo, language and syntax editing would make the paper read more clearly in a few places.

A: Thank you for pointing out these issues to our attention. They have been addressed in the revised version of the manuscript.

Sincerely,

Dr. Yizi Chen

---

## [Decision Letter · Decision Letter 1]

22 Jan 2024

Automatic vectorization of historical maps: a benchmark

PONE-D-23-22523R1

Dear Dr. Yizi,

We’re pleased to inform you that your manuscript has been judged scientifically suitable for publication and will be formally accepted for publication once it meets all outstanding technical requirements.

Kind regards,

Yawen Lu, Ph.D

Academic Editor

PLOS ONE

Additional Editor Comments (optional):

Dear Dr. Yizi:

PONE-D-23-22523R1

Congratulations! Based on the reviews and recommendations of reviewers, I am pleased to inform you that the above paper has been ACCEPTED for publication on PLOS One with no further changes.

Reviewers' comments:

Reviewer's Responses to Questions

**Comments to the Author**

1. If the authors have adequately addressed your comments raised in a previous round of review and you feel that this manuscript is now acceptable for publication, you may indicate that here to bypass the “Comments to the Author” section, enter your conflict of interest statement in the “Confidential to Editor” section, and submit your "Accept" recommendation.

Reviewer #3: All comments have been addressed

2. Is the manuscript technically sound, and do the data support the conclusions?

Reviewer #3: Yes

3. Has the statistical analysis been performed appropriately and rigorously? 

Reviewer #3: Yes

4. Have the authors made all data underlying the findings in their manuscript fully available?

Reviewer #3: Yes

5. Is the manuscript presented in an intelligible fashion and written in standard English?

Reviewer #3: Yes

6. Review Comments to the Author

Reviewer #3: This work provides a comprehensive benchmark for the vectorization of historical maps. The benchmark is comprehensive, well-designed, and properly evaluates multiple variants for historical map vectorization. It advances the field through extensive experiments and the identification of an improved vectorization pipeline. The availability of the data and models is also a valuable contribution.

7. PLOS authors have the option to publish the peer review history of their article (what does this mean?). If published, this will include your full peer review and any attached files.

Reviewer #3: No

---

## [Editor Report · Acceptance letter]

7 Feb 2024

PONE-D-23-22523R1 

PLOS ONE

Dear Dr. Chen, 

I'm pleased to inform you that your manuscript has been deemed suitable for publication in PLOS ONE. Congratulations! Your manuscript is now being handed over to our production team.

Kind regards, 

on behalf of

Dr. Yawen Lu 

Academic Editor

PLOS ONE